# Transcranial Direct Current Stimulation Effects on the Neural Substrate of Conceptual Representations

**DOI:** 10.3390/brainsci13071037

**Published:** 2023-07-07

**Authors:** Sonia Bonnì, Ilaria Borghi, Michele Maiella, Elias Paolo Casula, Giacomo Koch, Carlo Caltagirone, Guido Gainotti

**Affiliations:** 1Non-Invasive Brain Stimulation Unit, Department of Behavioral and Clinical Neurology, Fondazione Santa Lucia IRCCS, 00179 Rome, Italy; s.bonni@hsantalucia.it (S.B.); i.borghi@hsantalucia.it (I.B.); m.maiella@hsantalucia.it (M.M.); elias.casula@hsantalucia.it (E.P.C.); g.koch@hsantalucia.it (G.K.); 2Section of Human Physiology, University of Ferrara, 44121 Ferrara, Italy; 3Department of Systems Medicine, University of Rome Tor Vergata, 00133 Rome, Italy; c.caltagirone@hsantalucia.it; 4Department of Clinical and Behavioral Neurology, Fondazione Santa Lucia IRCCS, 00179 Rome, Italy; 5Institute of Neurology, Catholic University of the Sacred Heart, 00168 Rome, Italy

**Keywords:** living and non-living categories, ATLs, taxonomic and thematic relations, TPJ, verbal and pictorial representations, tDCS

## Abstract

The aim of this study was to shed light on the neural substrate of conceptual representations starting from the construct of higher-order convergence zones and trying to evaluate the unitary or non-unitary nature of this construct. We used the ‘Thematic and Taxonomic Semantic (TTS) task’ to investigate (a) the neural substrate of stimuli belonging to biological and artifact categories, (b) the format of stimuli presentation, i.e., verbal or pictorial, and (c) the relation between stimuli, i.e., categorial or contextual. We administered anodal transcranial direct current stimulation (tDCS) to different brain structures during the execution of the TTS task. Twenty healthy participants were enrolled and divided into two groups, one investigating the role of the anterior temporal lobes (ATL) and the other the temporo-parietal junctions (TPJ). Each participant underwent three sessions of stimulation to facilitate a control condition and to investigate the role of both hemispheres. Results showed that ATL stimulation influenced all conceptual representations in relation to the format of presentation (i.e., left-verbal and right-pictorial). Moreover, ATL stimulation modulated living categories and taxonomic relations specifically, whereas TPJ stimulation did not influence semantic task performances.

## 1. Introduction

Semantic disorders, one of the most important symptoms of vascular and degenerative forms of aphasia, can be considered a pathology of ‘semantic memory’, which was defined by Tulving [1] as a shared conceptual knowledge that builds over time and through multimodality experiences of concepts.

The classical notion, assuming that there may be a unitary localizable substrate for the representation of concepts (e.g., refs. [2,3]), was recast by Damasio’s [4,5] dynamic construct of higher-order convergence zones. This construct assumed that there is no single localizable store for a given entity’s meaning and that concept retrieval results from a dynamic process of recollection of modality-specific bits of memories are stored near the sensory portals and motor output sites of the system. Even though all contemporary models of conceptual representations draw upon the construct of higher-order convergence zones, some of these models support the unitary nature of these convergence zones, whereas other models assume the existence of multiple higher-order convergence zones. These multiple convergence zones could result from dichotomies between different (verbal vs. non-verbal) formats of semantic representations, different (living vs. non-living) categories of knowledge, or different (taxonomic vs. thematic) relations that can exist between concepts.

The first model, assuming the existence of multiple higher-order convergence zones and supporting different kinds of semantic representations, was proposed by Warrington and co-workers [6,7,8,9]. These authors argued that, even though all conceptual representations are based on integrations among various sensory modalities, the ‘weighting’ of these modalities should be different for distinct conceptual categories. Therefore, conceptual representations based on distinct sources of knowledge should have different neuroanatomical substrates, being bound to the cortical projections of the sensory-motor systems that play a crucial role in the construction of each conceptual category.

In keeping with these predictions, Warrington et al. [7,8,9] found that the anatomical locus of the lesion was different in patients with category-specific disorders for living beings and manmade objects. In particular, Warrington and Shallice [8] described the selective impairment of the categories of animals and plant life in four patients with herpes simplex encephalitis/HSE involving the anterior temporal lobes (ATLs), whereas Warrington and McCarthy [7,9] reported on the prevalent impairment of tools and other manmade objects in two vascular patients with lesions affecting the left fronto-parietal cortices.

This evidence was confirmed with surprising regularity by other authors, who investigated the etiology of lesions, the pattern of cognitive impairment, and the distribution of the anatomical lesions associated with category-specific semantic disorders for biological entities (e.g., refs. [10,11,12,13,14,15,16]) or for tools and other manmade objects (e.g., refs. [17,18,19]; see [20,21] for general reviews). Gainotti [20] explained these data arguing that the representation of biological entities relies on ATLs, where the ventral stream of visual processing converges with auditory, olfactory, and gustatory inputs. On the other hand, the fronto-parietal, sensory-motor cortices play a major role in the representation of artifacts because the dorsal stream of visual processing converges in these areas with body-related and action-oriented structures. A unitary account of the neural substrate of semantic knowledge was, on the contrary, proposed by the ‘hub and spoke’ model of semantic knowledge [22,23,24], based on clinical data, i.e., the severe and selective defect of semantic memory observed in patients with semantic dementia (SD). Since this disease is due to focal atrophy of the ATLs [25], the ‘hub and spokes’ model maintains that, in addition to different sensory, motor, and language processing areas (‘spokes’), the neural network for semantic memory requires a single convergence zone (‘hub’), located in both ATLs activating interactive semantic representations in all modalities and for all semantic categories.

Neuroimaging data supporting both the ‘category-specificity’ and the ‘hub and spokes’ model have been reported by several authors. Data in agreement with the different neuroanatomical lesions of patients showing a category-specific impairment for biological entities and manmade objects were obtained in a series of neuroimaging studies (refs. [26,27,28,29,30]; see [31] for general review). On the other hand, several fMRI (e.g., refs. [32,33]), TMS (e.g., refs. [34,35]), and ECog (e.g., ref. [36]) studies have supported the role of the ATL in semantic cognition (see [24] for general review).

A second controversy about the neural substrate of conceptual representations concerns the ‘format’ of representations subtended by the left and right ATLs. The original version of the ‘hub-and-spokes’ model [22,23,37] assumed that conceptual representations should be ‘stored’ in the same ‘amodal’ format in the right and left ATLs, whereas Snowden et al. [38,39,40] and Gainotti [41,42] maintained that the format of these representations should be verbal in the left ATL and non-verbal (pictorial) in the right ATL. The ‘unitary’ ‘hub-and-spokes’ model was based on neuropsychological data of patients with semantic dementia (SD) in which bilateral atrophy of the ATLs can lead to a semantic impairment affecting all modalities of concepts comprehension and production. However, Snowden et al. [38,39,40] and Gainotti [41,42] argued that semantic impairment is specifically ‘multi-modal’ in moderate-to-advanced stages of SD when atrophy affects the ATLs bilaterally. In contrast, in the early stages of SD, atrophy can be more lateralized. In these patients, semantic impairment mainly affects lexical-semantic knowledge when atrophy prevails on the left side and non-verbal representations when the atrophy prevails on the right side. Therefore, Gainotti [41,42] hypothesized that conceptual representations might be stored in a verbal format in the left ATL and a pictorial format in the right ATL. A different interpretation of these data was offered by Lambon Ralph et al. [24] and Lambon Ralph [43], who claimed that multimodal semantic representations are jointly subsumed by both left and right ATLs, but a graded specialization might emerge as a consequence of differential connectivity of the ‘hub’ with primary language, sensorimotor, or limbic regions. In other words, Lambon Ralph et al. [24] and Lambon Ralph [43] assumed that the difference between the semantic representations subsumed by the right and left ATL was quantitative and non-qualitative and due to the connectivity of the ‘hub’ with language areas on the left side and visual-perceptual areas on the right side of the brain.

Neuroimaging and non-invasive brain stimulation studies tried to check these alternative models concerning the format of semantic information processed by the right and left ATLs. Neuroimaging data supporting the different formats of semantic information processed by the right and left ATLs have been obtained by some authors, e.g., refs. [44,45,46], whereas functional neuroimaging data supporting the existence of brain regions representing amodal conceptual knowledge have been reported by other authors (e.g., refs. [32,33,47]). Furthermore, neuromodulation studies consistent with the ‘hub-and-spoke’ model were obtained from healthy subjects by some authors (e.g., refs. [34,35]), whereas other authors (e.g., ref. [48]) showed that the modulation elicited by rTMS during picture naming in healthy subjects was greater for the left than the right ATL. Moreover, Bonnì et al. [49], using continuous theta-burst stimulation (cTBS), supported the hypothesis that conceptual representations are stored in a different format (verbal and pictorial) in the right and left ATLs, even though paradoxical results were obtained.

The last controversy regarding the neural substrate of conceptual representations concerns the distinction between two kinds of relations that can exist between concepts, namely (a) categorical/taxonomic relations based on intrinsic similarity (i.e., shared features) and (b) contextual/thematic relations based on extrinsic relations between two objects. Some authors (e.g., refs. [50,51,52,53,54,55]) maintain that only taxonomic relations are supported by ATLs, whereas associative/thematic relations are underpinned by temporo-parietal junction (TPJ) cortices. On the other hand, other authors (e.g., refs. [56,57,58]) maintain that the processing of conceptual similarity and thematic relations rely upon the same semantic network centered on ATLs.

A relevant overlap could be noted, from a cognitive and anatomical perspective, between the taxonomic–thematic dichotomy and the contrast between biological and artifact categories proposed by Warrington et al. [6,7,8,9]. Indeed, taxonomic relations and biological categories recognition are based on visual features allowing object identification subsumed by ATLs, whereas thematic relations and manmade object knowledge are based on body-related and action-oriented dorsal (fronto-parieto-temporal) areas.

In the present study, we applied anodal tDCS over ATLs and TPJs cortices during the execution of a semantic task (the ‘Thematic and Taxonomic Semantic/TTS/task’ [59]) to simultaneously investigate (a) the distinction between stimuli presented in a verbal and a pictorial format, (b) living and non-living categories, and (c) those connected by a taxonomic/categorical or by a thematic/contextual relation.

Non-invasive transcranial stimulation techniques, such as tDCS, might contribute to clarifying the role of ATLs and TP cortices in conceptual representations, allowing us to test the involvement of different cerebral areas in cognitive task execution.

Specifically, the right and left cortices of ATL and TP were targeted to test the hemispheric lateralization of verbal and pictorial representations. At the same time, ATL and TP cortices were targeted to test the hypothesis assuming that only taxonomic knowledge should be supported by ATLs, whereas associative/thematic knowledge should be underpinned by TP cortices. Finally, the role of ATLs in the representation of living entities and the role of more dorsal parietal cortices in the representation of artifacts was investigated.

Notably, anodal tDCS is usually associated with an enhancement of the stimulated brain area, whereas cathodal tDCS is often described as an inhibitory brain stimulation method. However, this is a canonical assumption exclusively in the motor domain, while it is rarely observed in the cognitive domain. In particular, semantic processing seems to be somehow ‘‘immune” to cathodal effects” [60,61]. On the contrary, many studies (reviewed by [62]) found an effect following anodal tDCS on semantic processing in healthy subjects, although these effects varied across studies in terms of enhancement (decreased response time) or disruptions (increased response time). This apparent inconsistency was clarified by Pisoni et al. [63], who showed that activation of an area involved in semantic processing could disrupt performances by increasing the competition among alternative stimuli, a construct labeled semantic interference (SI) effect by Rosinski [64].

Pisoni et al. [63] showed that anodal tDCS over the left superior temporal gyrus (STG) significantly augmented the SI effect, increasing the competition between category exemplars, whereas anodal tDCS over the left inferior frontal gyrus led to a reduction of the same effect. In line with these results, Pisoni et al. [65] investigated the construct of SI with a proper name retrieval task by means of tDCS. These authors found that anodal tDCS over the left ATL decreased naming accuracy compared to sham stimulation because participants produced significantly more intrusions. They argued that stimulation may have increased interference among arising competitors when retrieving the correct name associated with the presented face, as indicated by the longer response latencies in the association task after real tDCS.

Since the task we used in the present study implied strong competition among alternative response stimuli, we expected a SI increase following the enhancement of excitability produced by anodal tDCS, together with a worsening of the corresponding performance.

The following predictions were therefore made: if multimodal semantic representations of living and non-living entities are jointly subsumed by both left and right ATLs, then only stimulations of these structures should be effective, worsening results irrespectively of the format (verbal vs) of the category of stimuli (living vs. non-living), and of the relations between related stimuli (thematic vs. taxonomic). On the contrary, if the neural substrate of conceptual representations is distinct for different categories of stimuli or for the format in which concepts are represented, then these category-related or format-related differences should be revealed by the effects of tDCS. Therefore, the neuromodulation should selectively concern stimuli belonging to living categories, presented in a verbal format when stimulating left ATL and pictorial format when stimulating right ATL. Analogously, if the ATLs mainly subsume taxonomic relations, whereas TPJ cortices underpin thematic relations, then ATL and TPJ stimulation should mainly influence the detection of taxonomic and thematic relations between conceptual representations, respectively.

## 2. Material and Methods

### 2.1. Participants

Twenty participants (M Age: 25.55 ± 2.41; 16 f/4 m) were recruited among undergraduate and graduate students during their internship at the Santa Lucia Foundation Hospital. All participants were right-handed, according to the Edinburgh Handedness Inventory [66]. All were native speakers of Italian with normal or corrected-to-normal vision, and they all provided written informed consent for their participation in the study. Following standard procedures in experiments involving tDCS [67], the participants were screened and excluded if they reported any psychiatric, psychological, or neurological disorder or if they reported brain injuries, migraines, epileptic seizures, or a family history of epilepsy. The choice to test 20 participants, divided into 2 groups, was based on the effect size reported by previous studies using tDCS and semantically guided tasks with a similar paradigm (e.g., refs. [68,69]). Specifically, the appropriateness of our sample size was established by a power calculation performed with G*Power software (American Psychological Association, Washington DC, USA), which indicated that 2 groups of 10 participants would be required to detect an effect with a power of 0.95 and an α level of 0.05. The experimental procedures were approved by the local ethical committee of the Santa Lucia Foundation, and all the experimental participants were conducted in adherence to the Declaration of Helsinki.

### 2.2. Experimental Procedure

Two different groups underwent three anodal tDCS sessions: one group (ATL group) was stimulated over the right ATL, left ATL, and vertex; the other group (TPJ group) was stimulated over the right TPJ, left TPJ, and vertex. A “wash-out” period of 1 week between sessions was planned to prevent unlikely after-effects of the stimulation.

### 2.3. tDCS Procedure

Study procedures and parameters are in accordance with safety and application guidelines for tDCS [52]. Stimulation was delivered by personnel trained in tDCS administration. The tDCS electrodes were applied to the scalp, and the electric current was delivered by a constant current stimulator using saline-soaked sponge electrodes (4 × 4 cm^2^). The active electrode (anode) was placed on the scalp according to the 10–20 International EEG System. The ATL group was stimulated over T8, T7, and CZ, corresponding to rATL, lATL, and vertex, and the TPJ group was stimulated over CP6, CP5, or CZ, corresponding to rTPJ, lTPJ, and vertex [68,70,71,72]. The vertex was chosen as the control area, given the evidence supporting that stimulation of the vertex does not affect the ongoing processes involved in task execution. (e.g., ref. [73]). Furthermore, stimulating a control site provides the same scalp sensation as tDCS stimulation to the targeted region. The reference (cathode) was placed over the contralateral shoulder during CP5, CP6, T7, and T8, while it was placed randomly over the right or left shoulder during the stimulation over the control site (CZ). During the stimulation, the current was increased to 2 mA and delivered for 20 min (with a 30 s ramp up and ramp down). All participants were asked to report the degree to which they experienced a list of side effects (headache, neck pain, scalp pain, tingling, stinging/itching, burning sensation, skin redness, drowsiness, concentration problems, or severe mood changes, e.g., ref. [74]). None of them reported major complaints or discomfort associated with stimulation.

### 2.4. Thematic and Taxonomic Semantic Task

#### 2.4.1. Stimuli

The thematic and taxonomic semantic (TTS) task was composed of stimuli triplets consisting of three pictures or words: a target item, an item semantically related to the target (hit), and an item unrelated to the target (distractor). Based on Gainotti and colleagues’ [31] TTS stimuli dataset, we created three lists of 40 stimuli triplets in pictorial or verbal modality, i.e., three pictorial (P1, P2, and P3) and three verbal (V1, V2, and V3) lists. Each list was balanced in order to generate triplets with stimuli belonging to the living (N = 20) or non-living categories (N = 20) having an exclusive taxonomic thematic relation (N = 20) or contextual semantic relation (N = 20). The stimuli lists are available online on the OSF registry (https://osf.io/cr9tz/?view_only=c56b00ea05ff43d78a4f076dab1f08f3; accessed on 5 December 2022).

#### 2.4.2. Task Procedure

Subjects performed the task in a silent room, sitting with their midsagittal plane aligned with the computer screen at a 70 cm distance. The E-prime 2 software (Psychology Software Tools, Pittsburgh, PA, USA) was used for the experimental procedure. Before the experimental procedure, participants were instructed about the stimulation and the tasks by means of a training session consisting of the presentation of five pictorial and five verbal triplets not used in the experimental phase. During the task, participants saw the triplets consisting of the target item on the top of the screen and two items below, one related to the target (hit), the other unrelated to the target (distractor). The triplets (pictures or words) were presented for a 3000 ms duration, followed by a 1500 ms period in which a fixation point was shown on the screen. The instruction was to indicate which of the two stimuli presented below was related to the target item. Subjects were asked to press, as quickly as possible, one of two response buttons on the keyboard that corresponded to the location of their selected item (left stimulus: left arrow or right stimulus: right arrow). In order to avoid a response bias and left/right asymmetry effects in processing semantic relationships [75,76], correct responses were equally and randomly presented in the right or the left position on the screen. Each experimental session consisted of the presentation of both pictorial and verbal lists, which were randomly administered with a two-minute interval in between. We used different pictorial and verbal stimuli for every session (Figure 1). Each session started with 10 min of tDCS, after which the subjects began the TTS, which lasted until the end of the stimulation. The tDCS was applied for 20 min, and the 10 min task was performed during the tDCS application [77]. The order of stimulation site (Vertex, rTPJ, and lTPJ or Vertex, rATL, and lATL), list (1, 2, and 3), and the format of stimuli (verbal and pictorial) were counterbalanced across the experimental condition. For each stimulation site, 40 triplets of items were administered and analyzed. Reaction times (RTs) and accuracy (percentage of correct response) were recorded using E-prime software 2 (Psychology Software Tool, Inc., Pittsburgh, PA, USA, www.psychtoolbox.org; accessed on 5 December 2022).

### 2.5. Statistical Analysis

All data were analyzed using SPSS version 22 (SPSS Inc., Chicago, IL, USA). The normal distribution of data was assessed by means of Shapiro–Wilks’ test. The level of significance was set at α = 0.05. The sphericity of the data was tested with Mauchly’s test; when sphericity was violated (i.e., Mauchly’s test < 0.05), the Huynh–Feldt ε correction was used. Behavioral performances in the TTS were evaluated in terms of accuracy (percentage of correct response) and RTs.

As a first step, we assessed the effect of “site of stimulation” (rATL, lATL, and CZ for the ATL group; rTPJ, lTPJ, and CZ for the TPJ group) over the cognitive performance when using two presentation formats, i.e., verbal and pictorial. To this aim, we conducted two one-way repeated-measure ANOVAs with factor site of stimulation, separately for verbal and pictorial material, for the two groups. As a second step, we assessed the same effect considering the category (i.e., living vs. non-living categories) and the (taxonomic vs. thematic semantic) relation between stimulus and response, with separate one-way ANOVAs. When a statistically significant effect was observed, paired *t*-tests, Bonferroni corrected, were used for post hoc analyses. The threshold of significance was set at *p* < 0.05.

## 3. Results

The analysis of ATL stimulation revealed a general effect of site of stimulation over accuracy in verbal presentation [F(2,27) = 3.442; *p* = 0.047; η2 = 0.203] and over RTs in pictorial presentation [F(2,27) = 3.827; *p* = 0.034; η2 = 221] (Figure 2A,C). Specifically, post hoc analysis revealed a significant difference in accuracy in the verbal presentation between lATL (83.5 ± 6.89%) and rATL (89.75 ± 3.21%) (*p* = 0.044) (Figure 2A) and a significant difference in RTs in the pictorial presentation between rATL (1416.45 ± 158.90 ms) and cz (1225.33 ± 161.22 ms) (*p* = 0.040) (Figure 2C).

The analysis of category variables showed an effect of the stimulation site on the living categories’ accuracy in the verbal presentation [F(2,27) = 11.03; *p* < 0.001; η2 = 0.450] and the living categories’ RTs in the pictorial presentation [F(2,27) = 3.795; *p* = 0.035 η2 = 0.219]. Post hoc analysis performed over living categories’ accuracy in the verbal presentation showed a significant difference of lATL compared to Cz (*p* = 0.007) and rATL (*p* < 0.001) (Figure 3A). Indeed, the stimulation over the lATL induced a worsening in accuracy performance (81.09 ± 7.73%) with respect to Cz (89.66 ± 4.43%) and rATL (92.59 ± 4.19%) stimulation. On the other hand, post hoc analysis of living categories’ RTs in the pictorial presentation showed only a trend to significance (*p* = 0.059) in the comparison between rATL (1401.56 ± 170.66 ms) and cz (1219.37 ± 137.93 ms) (Figure 3B).

The analysis of semantic relation showed an effect of the stimulation site on the taxonomic relation’s accuracy in the verbal presentation [F (2,27) = 3.974; *p* = 0.031; η2 = 0.227] (Figure 3C). Post hoc analysis revealed a significant difference in rATL compared to lATL (*p* = 0.027). Indeed, the stimulation over rATL induced an increase in accuracy (87.5 ± 5.40%) when compared to lATL (76.47 ± 12.21%) stimulation.

Moreover, the analysis of semantic relation showed an effect of the stimulation site for the taxonomic relation’s RTs in the pictorial presentation [F (2,27) = 4.834; *p* = 0.016; η2 = 0.264] (Figure 3D). Post hoc analysis revealed a significant difference in rATL compared to Cz (*p* = 0.027) and lATL (*p* = 0.05). Indeed, the stimulation over rATL induced a slowdown in RTs (1464.88 ± 141.31 ms) when compared to cz (1253.10 ± 173.33 ms) and lATL (1273.08 ± 186.64 ms) stimulation.

The analysis of TPJ stimulation did not show a general effect of SITE stimulation (*p* > 0.05) either on semantic relation or on category variables.

## 4. Discussion

The main results of the present investigation can be summarized as follows: (a) ATL stimulation had a strong and consistent influence on various aspects of conceptual representations, whereas TPJ stimulation had only a marginal and inconsistent influence; (b) the influence of ATL stimulations was not homogeneous across criteria of evaluation (accuracy vs. RTs) of SITE stimulation effect on the format of representations (verbal vs. pictorial), categories (living vs. non-living), and semantic relation (taxonomic vs. thematic) variables.

However, these non-homogeneous results remained logically consistent. Thus, the SITE stimulation effect consisted of a worsening in accuracy performance for the verbal presentation of living categories after stimulation of the lATL and a slowdown in RT for the pictorial presentation of the same categories after stimulation of the rATL. Both these results are consistent with models assuming that the ATLs play a critical role in the representation of biological entities (e.g., refs. [8,20,78]) and that these representations are mainly ‘stored’ in a verbal format in the left ATL and a non-verbal pictorial format in the right ATL (e.g., refs. [38,39,40,41,42,54]). Furthermore, the analysis of RTs in the pictorial presentation showed a worsening after stimulation of rATL for the taxonomic relations and the living categories. The first result is consistent with models assuming that pictorial representations may be subsumed by the right ATL and that taxonomic relations, mainly based upon a convergence of perceptual features, may be preferentially subsumed by the ATLs, where these features converge (e.g., refs. [54,55,79]). The second result confirms that the right ATLs play a greater role in the pictorial representation of biological entities, in agreement with the leading role of the ATLs in the representation of living entities and with their pictorial format in the right ATL.

On the other hand, no evidence was found in support of the hypothesis that TPJ cortices may underpin thematic relations. The marginal influence of TPJ stimulation on conceptual representations was only suggested by the reduction of accuracy between the control site and rTPJ found in the verbal presentation of the non-living categories. Indeed, several authors (e.g., refs. [7,9,20,78,79,80]) have suggested that the parietal areas may play an important role in the representation of tools. However, it must be noted that, according to these models, a critical role in this function should be played by the left fronto-parietal cortices rather than by the right TP cortices. Therefore, the results obtained through TPJ stimulation do not seem very relevant in disentangling the neural substrate of conceptual representations.

Despite the non-homogeneity of our results, they are consistent with our predictions from both the methodological and theoretical points of view. In agreement with the implications derived from the construct of the SI effect, we confirmed a worsening of performance as an effect of anodal tDCS when there is strong competition among the alternative response items. With respect to the neural substrate of conceptual representations and the predictions based on the unitary (‘hub-and-spokes) or on the multiple ‘higher-order convergence zones’, our results are mixed because they have shown that the role of the ATLs is greater for living entities in comparison to artifacts and that the left ATL is preferentially involved in verbal (and the right ATL in non-verbal) representations. The dissociation observed between living and non-living entities (obtained only through the modulation of accuracy in the verbal presentation and of RTs in the pictorial presentation) is at variance with the predictions of the ‘hub-and-spokes’ model, whereas the preferential involvement of the left ATL in verbal and of the right ATL in pictorial representations is consistent both with the ‘ different format’ and the ‘graded specialization in semantic processes’ between the left and right ATLs hypothesis proposed by Lambon Ralph et al. [24,46] and Rice et al. [81]. These general interpretations of results obtained in our tDCS study are consistent with those of recent anatomo-clinical investigations, which have tried to evaluate the category-specificity and the format of semantic disturbances observed in patients with semantic dementia or semantic variant primary progressive aphasia. Indeed, Chan et al. [82], Lisbon et al. [83], and Henderson et al. [84] reported a disproportionate impairment of living beings in most patients with semantic variant primary progressive aphasia and involvement of the left ATL, whereas Chan et al. [82], Chen et al. [85], and Guo et al. [86] showed that semantic processing is orchestrated through interactions between a critical ATL hub and lateralized modality-selective processing nodes.

## 5. Study Limitations

The non-homogeneous results of our study could be, at least in part, due to methodological limitations. Indeed, some studies (e.g., refs. [87,88]) showed that tDCS effects could be small or negative, especially for single sessions in healthy participants. Furthermore, some authors showed that tDCS effects are not always linear and attributed this confounding characteristic to different factors such as the initial activation state of the stimulated area (e.g., refs. [87,89]), the strength and duration of the administered current (e.g., refs. [67,90,91,92,93]), and even the characteristics of the task under investigation (e.g., its type and level of difficulty [94]).

A further limitation of our study could reside in the lack of control of the strength of relations existing between targets and the thematically or taxonomically related stimuli used in the ’TTS’ task. The reason for this was that norms of taxonomic and thematic relations similar to those used by Landrigan and Mirman [95] could be produced only for word pairs, whereas stimuli triplets composing our ‘TTS’ task were made both of words and pictures.

## 6. Conclusions

Taken together, the results of the present investigation are consistent with models assuming that different ‘higher-order convergence zones’ may subsume different conceptual categories and their verbal or non-verbal representations, whereas they do not support models surmising that different neural substrates may underpin distinct semantic (thematic and taxonomic) relations. Future investigations are necessary to confirm the preliminary conclusions that can be drawn from this study.

## Figures and Tables

**Figure 1 brainsci-13-01037-f001:**
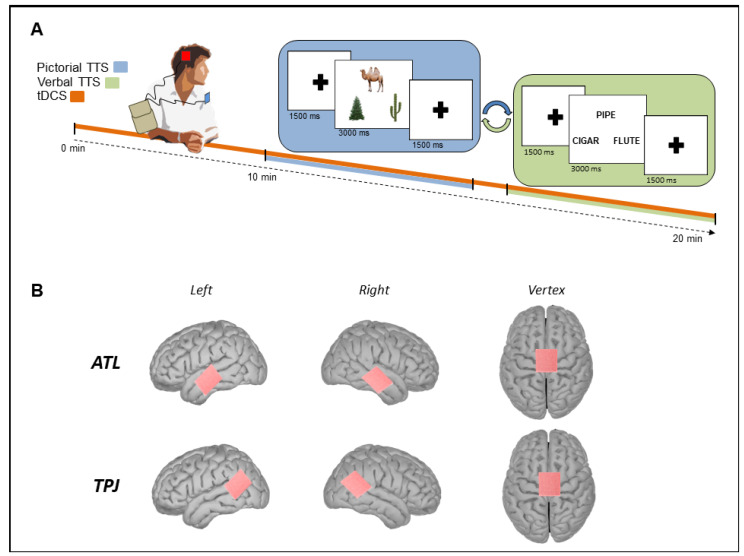
Panel (**A**). Experimental session design. The session starts with tDCS delivered on the target area (orange line); TTS begins after 10 min and lasts until the session’s end. Subjects performed both a pictorial (blue line) and a verbal (green line) version of the TTS randomly ordered, with a two minute interval between them. The whole session lasts around 20 min. Panel (**B**). tDCS electrode positioning. Representation of the electrode position for the anodal stimulation. First line represents the ATL group stimulation sites (from left: ATL left, ATL right, and vertex). Bottom line represents the TPJ group stimulation sites (from left: TPJ left, TPJ right, and vertex).

**Figure 2 brainsci-13-01037-f002:**
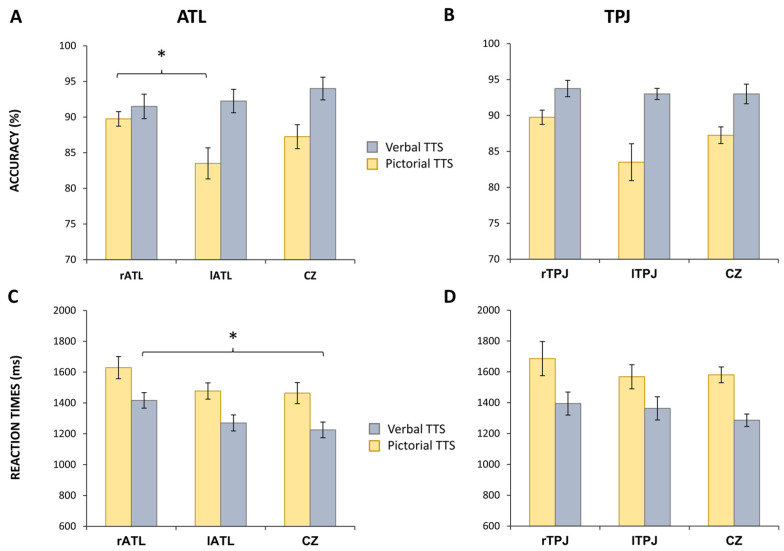
Effects of anodal tDCS on verbal and pictorial TTS. Panels (**A**,**C**) show accuracy and reaction times for verbal (green) and pictorial (blue) TTS in the anterior temporal lobe (ATL) groups. Panels (**B**,**D**) show accuracy and reaction times in verbal and pictorial TTS in temporo-parietal junction (TPJ) groups. Panel (**A**) shows that tDCS on lATL worsens the accuracy in verbal TTS compared to rATL. Panel (**C**) shows that tDCS on rATL slows down reaction times in pictorial tasks compared to lATL and Cz. Abbreviations: TTS, thematic and taxonomic semantic task; rATL, right ATL; lATL left ATL; Cz, vertex; rTPJ, right TPJ; lTPJ, left TPJ. * = *p* ≤ 0.05.

**Figure 3 brainsci-13-01037-f003:**
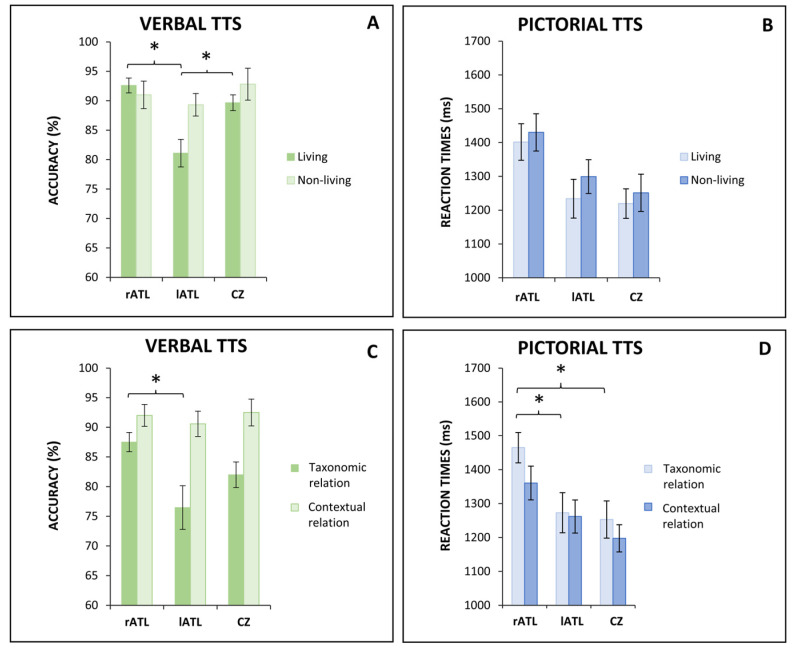
Effects of anodal tDCS on semantic variables. Panel (**A**) shows that tDCS on lATL worsens the accuracy in verbal living categories (dark green) compared to rATL and CZ. Panel (**B**) shows that tDCS on rTPJ worsen the accuracy in non-living categories (light green) compared to CZ. Panel (**C**) shows that tDCS on rATL slows down the reaction times in the pictorial living categories (light blue) compared to lATL and CZ. Panel (**D**) shows that the tDCS on rATL slows down the reaction times for the pictorial taxonomic relation (light blue) compared to lATL and CZ. Abbreviation: TTS, thematic and taxonomic semantic task; rATL, right anterior temporal lobe; lATL, left anterior temporal lobe; CZ, vertex; rTPJ, right temporo-parietal junction; lTPJ, left temporo-parietal junction: * = *p* ≤ 0.05.

## Data Availability

https://osf.io/cr9tz/?view_only=c56b00ea05ff43d78a4f076dab1f08f3; accessed on 5 December 2022.

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
