# Peer review of "Transcranial Direct Current Stimulation Effects on the Neural Substrate of Conceptual Representations"

_brainsci, 2023, doi:10.3390/brainsci13071037_

Round 1

Reviewer 1 Report

Comments and Suggestions for Authors

Introduction: Instead of referring to ‘non-verbal (pictorial)’, would it be preferable to refer to ‘visual (pictorial)’ in keeping with Warrington’s distinction? This occurs in several locations.

Results: In Figure 3, it is not clear why Verbal TTS reports accuracy and not RTs and why Pictorial TTS reports RTs and not accuracy. Please explain.

Comments on the Quality of English Language

Some minor writing edits are needed where there are extraneous articles (the) or missing plurals (zones) in various places throughout the manuscript.

Line 193 typo for ‘to’

Line 213 stimulation ‘was’ delivered

Reviewer 2 Report

Comments and Suggestions for Authors

This article investigated shed light on the neural substrate of conceptual representations. It is confirmed that ATL stimulation influenced all conceptual representations in relation to format of presentation and ATL stimulation modulated specifically living categories and taxonomic relations, whereas TPJ stimulation did not influence the semantic task performances. The article is well organized and its presentation is good, which deserve publication in brain sciences. Some minor concerns are listed below.

1.     Figure 1 is not mentioned in the main text.

2.     The author citation format of the reference is not consistent; The year of reference 57 is not bold; The page number of reference 87 is not marked; The references are relatively outdated, and it is recommended to cite recent literature.

3.     Please add a paragraph about the limitation of the study.

Comments on the Quality of English Language

Minor editing of English language required

Reviewer 3 Report

Comments and Suggestions for Authors

This article is very interesting. Here are my suggestions attached. Best regards.
